# Relationships between Virulence Genes and Antibiotic Resistance Phenotypes/Genotypes in *Campylobacter* spp. Isolated from Layer Hens and Eggs in the North of Tunisia: Statistical and Computational Insights

**DOI:** 10.3390/foods11223554

**Published:** 2022-11-08

**Authors:** Manel Gharbi, Selim Kamoun, Chaima Hkimi, Kais Ghedira, Awatef Béjaoui, Abderrazak Maaroufi

**Affiliations:** 1Group of Bacteriology and Biotechnology Development, Laboratory of Epidemiology and Veterinary Microbiology, Institut Pasteur de Tunis, University of Tunis El Manar (UTM), Tunis 1002, Tunisia; 2Laboratory of Bioinformatics, Biomathematics and Biostatistics, Institut Pasteur de Tunis, University of Tunis El Manar (UTM), Tunis 1006, Tunisia

**Keywords:** *Campylobacter*, antibiotic resistance, virulotyping, Virulence-AR association, data analysis

## Abstract

Globally, *Campylobacter* is a significant contributor to gastroenteritis. Efficient pathogens are qualified by their virulence power, resistance to antibiotics and epidemic spread. However, the correlation between antimicrobial resistance (AR) and the pathogenicity power of pathogens is complex and poorly understood. In this study, we aimed to investigate genes encoding virulence and AR mechanisms in 177 *Campylobacter* isolates collected from layer hens and eggs in Tunisia and to assess associations between AR and virulence characteristics. Virulotyping was determined by searching 13 virulence genes and AR-encoding genes were investigated by PCR and MAMA-PCR. The following genes were detected in *C. jejuni* and *C. coli* isolates: *tet*(O) (100%/100%), *bla*_OXA-61_ (18.82%/6.25%), and *cmeB* (100%/100%). All quinolone-resistant isolates harbored the Thr-86-Ile substitution in GyrA. Both the A2074C and A2075G mutations in 23S rRNA were found in all erythromycin-resistant isolates; however, the *erm*(B) gene was detected in 48.38% and 64.15% of the *C. jejuni* and *C. coli* isolates, respectively. The machine learning algorithm Random Forest was used to determine the association of virulence genes with AR phenotypes. This analysis showed that *C. jejuni* virulotypes with gene clusters encompassing the *racR*, *ceuE*, *virB11*, and *pldA* genes were strongly associated with the majority of phenotypic resistance. Our findings showed high rates of AR and virulence genes among poultry *Campylobacter*, which is a cause of concern to human health. In addition, the correlations of specific virulence genes with AR phenotypes were established by statistical analysis.

## 1. Introduction

The most prevalent cause of bacterial gastroenteritis, *Campylobacter* spp., accounts for 5–14% of all diarrheal diseases worldwide [1]. Among the human-associated *Campylobacter* species, 95% of campylobacteriosis is caused by the *C. jejuni* and *C. coli* species [2], causing 96 million cases of diarrhea each year globally. In addition, *Campylobacter* is the second most prevalent agent of diarrhea in Europe (after *Salmonella*) [3]. *Campylobacter* was the second aetiological agent of outbreaks related to food and water poisoning in 2018 [4]. Contrary to European and developing countries, there are few reports of human campylobacteriosis in North African countries, including Tunisia, presumably owing to the low prevalence of the disease or the sporadic cases of infections. Globally, high rates of antimicrobial resistance (AR) have been increasingly noted, which dramatically reduced treatment options for campylobacteriosis. The National Antimicrobial Resistance Surveillance System (NARMS, Atlanta, GA30329, USA, 2019) reports that *Campylobacter* causes 448,400 illnesses that are resistant to treatment with antibiotics each year, along with an estimated 70 fatalities. Tetracyclines, macrolides, and fluoroquinolones are the main antibiotics used to treat *Campylobacter* infections [2]. Owing to its broad antimicrobial spectrum, ciprofloxacin (a fluoroquinolone) use in the food-producing animal sectors is common. However, fluoroquinolone-resistant *Campylobacter* isolates from both human and animal sources have been dramatically described during the last decades [5,6]. Additionally, macrolides (such as tilmicosin, tulathromycin, and tildipirosin) have been widely utilized in animals raised for food in many geographic regions [7,8], which has led to the selection and spread of *Campylobacter* isolates that are resistant to these antibiotics [9,10,11].

Tetracycline is a broad-spectrum antibiotic with low cost and high efficacy; therefore, it has been extensively used in animal farming [12]. However, similarly to other antimicrobial agents, high rates of tetracycline-resistant *Campylobacter* isolates from livestock have been reported worldwide [13,14]. Overall, in recent decades, high frequencies of resistance to tetracycline-, ciprofloxacin-, and erythromycin have been reported in *Campylobacter* strains [15]. Interestingly, as a result of antimicrobial resistance selection under therapeutic treatment or antimicrobial use as a growth promoter, the rates of multi-drug resistant (MDR) *Campylobacter* isolates have drastically increased in human medicine [16] and livestock [17,18]. Pork and poultry or poultry products are the main origins of *Campylobacter* spp causing human diseases; therefore, the potential of MDR isolates spreading from animals to humans is a real cause of concern for human health.

The C257T mutation in *gyrA* in the *Campylobacter* species is the most prominent mechanism mediating quinolones and fluoroquinolones resistance [19]. In various bacteria species, three molecular mechanisms encoding tetracycline resistance have been reported: (i) efflux pumps, (ii) ribosome target protection, and (iii) the enzymatic modification of the antibiotic, mediated by more than 60 genes [20,21,22]. The *tet*(O) gene is the dominant tetracycline resistance determinant that has been detected in the *Campylobacter* species [12,23,24,25]. The Tet(O) protein mediates resistance by removing tetracycline from its major binding site on the ribosome. In *Campylobacter*, the primary molecular mechanisms of macrolides resistance are changes in the ribosomal target and active efflux. The alteration of the ribosomal target can occur either by the enzymatic methylation of the region V of 23S rRNA or by point mutation in the ribosomal proteins L4 (*rplD* gene) and L22 (*rplV* gene) [26]. The *CmeABC* multidrug efflux pumps mediate the active efflux of the antibiotic [27]. Macrolide resistance mediated by rRNA methylation, encoded by the *erm*B gene, was firstly reported in *C. rectus* (1995) and currently is sporadically reported in *C. coli* and *C. jejuni* [28].

It is yet unclear whether a rise in AR in *Campylobacter* has enhanced this bacterium’s potential for pathogenicity or vice versa. There is currently no consensus among scientists about the relationship between AR and pathogenicity [29]. As a result, it is unclear if an increase in AR leads to an increase in genes encoding virulence factors in pathogenic bacteria like *Campylobacter*. AR acquisition is essential for bacteria to survive in environments rich with antibiotics, while the virulence genes are necessary to surmount the host defense systems [29]. Additionally, the acquisition of antimicrobial encoding genes may be linked to a reduction in virulence, while some data imply the opposite, that AR may improve or enhance virulence [30]. When bacteria are found in an environment with antibiotics, they may be able to increase their virulence by using virulence determinants to escape the host’s defenses throughout the host–pathogen interaction, suggesting the potential for pathogenicity enhancement [29,31]. Additionally, according to some studies, acquired resistance mechanisms include a fitness cost, which may reduce pathogenicity in bacteria, making them less aggressive when fighting host defense [30,32]. However, there is evidence that AR genes can be suppressed without any biological costs, while other adaptive features are produced without affecting virulence [32]. Owing to these facts, it appears that the acquisition of AR is required to allow harmful bacteria like *Campylobacter* to avoid antimicrobial therapy without compromising their virulence.

This study sought to determine whether specific virulence genes, resistance genes, and AR characteristics were associated with one another in *Campylobacter* isolates. To achieve this, we determined the antimicrobial susceptibility and investigated by PCR-selected genes of virulence and AR in a collection of *Campylobacter* isolates collected from laying hens and eggs. The relationship between the different aforementioned traits was then investigated using a variety of statistical and computational methodologies.

## 2. Materials and Methods

### 2.1. Ethics Statement

The Biomedical Ethics Committee of the Institut Pasteur de Tunis gave its approval to this study, and the sampling protocol was performed according to internationally recognized guidelines ISO 10272-1:2006 (Annex E) for the detection of *Campylobacter* spp. [33].

### 2.2. Bacterial Strains

One hundred seventy-seven *Campylobacter* isolates have been reported previously [34]. These isolates include 124 *C. jejuni* and 53 *C. coli* recovered from five laying hen farms located in the northeast of Tunisia between October 2017 and May 2018.

### 2.3. Antimicrobial Susceptibility Testing

For all isolates, antimicrobial susceptibility testing was performed by the disk diffusion method on Mueller–Hinton medium (Bio Life, Milan, Italy) according to the European Committee on Antimicrobial Susceptibility Testing (EUCAST, City, Country, 2017) guidelines [2]. The used antibiotics were (Oxoid, Basingstocken, UK): ampicillin (AMP,10 μg), amoxicillin/clavulanic acid (AMC, 10/20 μg), gentamicin (GEN, 10 μg), tetracycline (TET, 30 μg), erythromycin (ERY, 15 μg), nalidixic acid (NAL, 30 μg), ciprofloxacin (CIP, 5 μg), and chloramphenicol (CHL, 30 μg) [35].

### 2.4. Detection of Genes Encoding Virulence Factors

PCR was used to detect 13 virulence genes specific to *C. coli* and *C. jejuni*: *flaA* (motility); *cadF*, *racR*, and *dnaJ* (cell adhesion); *pldA*, *virB11*, and *ciaB* (colonization and invasion); *ceuE* (iron absorption system); *cdtA.B.C* (production of cytotoxins); *wlaN* and *cgtB* (expression of Guillain-Barré syndrome) (Table A1). Positive control strains from our collection were used in every PCR analysis [36].

### 2.5. PCR Detection of Genes Encoding AR

Fluoroquinolone resistance is commonly encoded by single point mutation (Thr-86-Ile) in the quinolone resistance-determining region (QRDR) of the GyrA subunit of the DNA gyrase enzyme [37]. For *C. jejuni* isolates, MAMA-PCR was performed as previously reported [38], while for *C. coli*, the used protocol was as cited by Zirnstein et al. [37]. MAMA-PCR was also used to detect point mutations at positions 2074 and 2075 in domain V of the 23S rRNA gene, which are related to erythromycin resistance, as described previously [39]. For all isolates, the following genes were detected by the classical PCR method: *erm*(B) (erythromycin resistance) Qin et al. (2014), *tet*(O) (tetracycline resistance), *aph*-3-1 (aminoglycosides resistance), *cmeB* (multidrug efflux pumps), and *bla*_OXA-61_ (beta-lactam resistance) (Table A2). Positive control strains from our collection were used in every PCR analysis [36].

### 2.6. Statistical Analysis

Statistical analysis was performed to investigate a possible association between virulence genes and AR in all isolates. We studied the antimicrobial susceptibility phenotypes (resistance/susceptibility) against the eight tested antibiotics (Amp, Amc, Cip, Nal, Ery, Tet, Chl, and Gen), and associated the latter with the presence/absence of the investigated virulence genes (*cadF*, *ciaB*, racR, *flaA*, *dnaJ*, *cdtA*, *cdtB*, *cdtC*, *virB11*, *pldA*, *wlaN*, *ceuE*, and *cgtB*). This was investigated first for all *Campylobacter* isolates, and then for the isolates of each species. The association test of each virulence gene with the AR phenotype was computed by Pearson’s chi-square or Fisher’s exact test using R software via RStudio (version 1.4.1103). Fisher’s exact test was used when the expected cell counts for the contingency table held less than five isolates. If the *p*-value < 0.05, the association was deemed statistically significant.

### 2.7. Network Generation

Two groups of networks were built connecting phenotypical AR with virulence genes, as well as AR genes with virulence genes. The networks were displayed via Cytoscape (https://cytoscape.org/) (20 February 2022) (version 3.8.1) (https://pubmed.ncbi.nlm.nih.gov/14597658/) (20 February 2022). These networks were built with the aim of revealing co-occurrence patterns and identifying interactions that could reveal information on the patterns of the incidence of virulence genes and AR across all *Campylobacter* isolates (only virulence genes that showed a statistically significant association were used to build the network).

### 2.8. Predictive Analysis Using Machine Learning Random Forest Algorithm

Following the statistical association test, a predictive analysis was performed using the machine learning Random Forest algorithm, via the randomForest R package (https://link.springer.com/article/10.1023%2FA%3A1010933404324) (20 February 2022), in order to determine the most important virulence genes that could be related with a specific AR phenotype. Classification trees are used in the analysis to establish, for each variable, its importance in classifying the data and determining the outcome through the production of an importance score [40]. Only virulence genes that showed a statistically significant association with AR through Pearson’s chi-square/Fisher’s exact test for all *Campylobacter* isolates (both *C. coli* and *C. jejuni* species together) were considered for this classification. The Random Forest measures the contribution of each virulence gene to a particular resistance phenotype. The algorithm produces a MeanDecreaseGini score that gives a valuable estimation of the significance of the variable in the model and thus, in our case, valuable information to determine which gene is more likely to be linked to an increased probability of a specific AR [41].

## 3. Results

### 3.1. Virulotypes and Phenotypic Profiling of AR

One-hundred-and-seventy-seven *Campylobacter* isolates (124 *C. jejuni* and 53 *C. coli*) were analyzed to determine the virulotype (content of genes encoding virulence factors) and phenotypic AR profiles. All isolates (*n* = 177, 100%) harbored the *flaA*, *cadF*, *ciaB*, and *cdt* genes, closely followed by the *racR* gene (*n* = 161, 90.96%) (Figure 1A). A close result was obtained when analyzing the 124 *C. jejuni* isolates. Indeed, the *flaA*, *cadF*, *ciaB*, and *cdt* genes were present in all isolates (100%), followed by the *dnaJ* (*n* = 119, 95.97%) and *ceuE* (*n* = 115, 92.74%) genes (Figure 1B). There were no discernible differences found in the *C. coli* species for the most common virulence genes. Indeed, all isolates contained the *flaA*, *cadF*, *racR*, *ciaB*, and *cdt* genes, whereas, the *pldA* gene was detected in 51 (96.22%) isolates. Interestingly, a major difference was observed concerning the *ceuE* gene, which was absent in all *C. coli* isolates but highly present in the *C. jejuni* ones (92.74%) (Figure 1C).

According to the phenotypic antimicrobial susceptibility profiling, all isolates were multi-drug-resistant, being resistant to at least three antibiotics belonging to different classes. Taking all the *Campylobacter* isolates, high rates of AR were observed for erythromycin (*n* = 175, 98.87%) and tetracycline (*n* = 174, 98.30%); however, a low resistance rate was observed for gentamicin (*n* = 2, 1.13%) (Figure 2A). When taken alone, the *C. coli* isolates showed a very high number of resistant isolates toward most of the antibiotics used except for ampicillin (*n* = 9, 16.98%) and gentamicin (*n* = 0) (Figure 2B). For the *C. jejuni* isolates, high resistance rates were detected for erythromycin (*n* = 122, 98.4%), tetracycline (*n* = 122, 98.4%), and chloramphenicol (*n* = 121, 97.6%); in contrast, the gentamicin resistance rate was low (*n* = 2, 1.61%) (Figure 2C).

### 3.2. Molecular Detection of AR Genes

All isolates (*n* = 177) carried the *tet*(O) and *cmeB* genes, according to the PCR data. In the β-lactam-resistant *C jejuni* and *C coli* isolates, the *bla*_OXA-61_ gene was found in 18.82% and 6.25%, respectively. For the quinolone-resistant isolates, the Thr-86-Ile mutation in GyrA was found in all *C. jejuni* and *C. coli* isolates. Similarly, all erythromycin-resistant isolates harbored the A2075G and A2074C mutations, while the *erm*(B) gene was detected in 53.71% (94/175) of the erythromycin-resistant *Campylobacter* isolates, being in 60 (48.38%) and 34 (60.15%) of the erythromycin-resistant *C. jejuni* and *C. coli* isolates, respectively. There was no isolate harboring the *aphA*-3 gene.

### 3.3. Statistical Analysis of Phenotypic AR with Virulence Genes

Pearson’s chi-square and Fisher’s exact tests were executed to study the association between the set of virulence genes and AR in all isolates showing resistance to 4–6 antibiotics, as well as those resistant to more than six antibiotics (Table 1). A significant correlation between AR and various virulence genes was observed, more specifically with *racR* [χ^2^ = 16.144, *p* = 5.871 × 10^−5^], *pldA* [χ^2^ = 3.8849, *p* = 0.04872], and *ceuE* [χ^2^ = 24.265, *p* = 8.393 × 10^−7^]. A similar analysis was also performed for isolates of each species. The *C. jejuni* isolates showed a significant relationship between AR and different virulence genes, and more precisely for *racR* [χ^2^ = 16.144, *p* = 16.144], *virB11* [χ^2^ = 8.2523, *p* = 0.004213], *pldA* [χ^2^ = 10.718, *p* = 0.001369] as well as *cgtB* [χ^2^ = 3.5443, *p* = 0.0933] (Table 2). However, no significant relationships were observed concerning the *C. coli* isolates (Table 3).

### 3.4. Network Analysis of Resistance, Virulence Genes, and Phenotypic AR

In order to examine the co-occurrence patterns, we generated networks describing the connections between (i) phenotypic AR with virulence genes and (ii) AR genes with virulence genes to provide information on the patterns and incidence of virulence genes and AR across all *Campylobacter* isolates. Figure 3 reveals three distinct networks that describe links between AR and the presence/absence of certain virulence genes for each isolate.

Focusing only on the virulence genes that showed a statistically significant association, we noticed the coexistence of some connections between phenotypic AR and specific virulence genes among some isolates more frequently than other ones. Approximately, for a third of the isolates (*n* = 50), there was a high frequency of connections linking nalidixic acid (Nal), tetracycline (Tet), erythromycin (Ery), ciprofloxacin (Cip), ampicillin (Amp), and chloramphenicol (Chl) resistance with the virulence genes *pldA* and *racR* (Figure 3A). Similarly, when looking into the networks generated for 100 antimicrobial-resistant isolates and the total number of *Campylobacter* isolates (*n* = 177), the same high-frequency connections were created between phenotypic AR and the virulence genes *pldA* and *racR* as shown in Figure 3B,C, respectively.

In Figure 4, we displayed three networks that show the connections between resistance genes and the virulence genes for each *Campylobacter* isolate. For 50 isolates, there was a high frequency of connections linking the following resistance genes: *cmeB*, *tet*(O), *Cj-gyrA*, and 23S rRNA (mutated) with the virulence genes *ceuE*, *pldA*, and *racR* and the *erm*B with *pldA* and *racR* (Figure 4A). However, for 100 and 177 isolates, the latter connections were conserved by adding new connections linking *erm*B with the virulence gene *ceuE*. New added links have shown a high frequency of connection between *bla*_OXA-61_ with *racR*, *pldA*, and *cgtB* and *Cc-gyrA* with *racR* and *pldA* (Figure 4B,C).

### 3.5. Predictive Analysis of AR/virulence Genes Links Using the Machine Learning Random Forest Algorithm

The Random Forest algorithm was used to further explore the possible association of the virulence genes that showed a significant association upon the statistical analysis for all *Campylobacter* isolates. In order to predict which one could be the best indicator of a specific AR, Random Forest produces a MeanDecreaseGini value, and the higher this value is, the higher the significance of the variable in the model.

This investigation showed that one virulence gene, *racR*, displayed the most important value with two antibiotics, nalidixic acid, and ciprofloxacin (Figure 5C,D). On the other hand, another gene, *ceuE*, has shown the most important value with five other antibiotics, Amoxicillin, Erythromycin, Tetracycline, Chloramphenicol, and Gentamicin (Figure 5A,E–H). Finally, the *pldA* gene showed an important value for Ampicillin only (Figure 5B).

## 4. Discussion

### 4.1. Antimicrobial Resistance and Corresponding Genotypes

The treatment of *Campylobacter* infections is currently jeopardized by the emergence of AR, which has become a complex challenge and a major issue for global public health. The Tunisian government lacks an integrated program for monitoring AR in primary human and production animal pathogens such as *C. jejuni*, *C. coli*, and *C. fetus*, making it difficult to implement new antimicrobial control and restriction measures. Furthermore, unlike other European countries, Tunisia has no specific legislation mandating campylobacteriosis testing. AR studies are thus critical for characterizing the circulating *Campylobacter* strains in Tunisian poultry. Mobile genetic elements, including plasmids and transposons, which can also carry virulence determinants, are highly associated with the global spread of AR. In Tunisia, research on AR in *Campylobacter* isolates from laying hens and eggs is scarce. Thus, herein, we analyzed 177 *Campylobacter* isolates (124 *C. jejuni* and 53 *C. coli*) from the layer hens and eggs collected in the north of Tunisia. The isolates were investigated to determine their virulotypes and AR phenotypes.

This research revealed no discernible differences in the status of certain virulence genes between *Campylobacter* isolates that are resistant to 4–5 antibiotics or to more than 6 antibiotics. Our findings revealed that resistance to erythromycin, tetracycline, quinolone, and ciprofloxacin is common, which can considerably restrict the number of the available treatment options of infections caused by such strains. High rates of resistance are anticipated because these antibiotics have been on the market for a long time and have been used widely in both legal and illegal situations.

The interaction(s) between AR and virulence is still poorly understood. However, there is strong scientific proof that the development of AR by the overexpression of genes encoding AR or multidrug-resistant efflux pumps causes a fitness cost to bacteria, such as lower growth rates and pathogenicity [39,42]. However, many other studies have found that pathogens’ acquisition of AR improves their fitness and virulence [30,43].

The majority of our isolates were resistant to tetracycline, ciprofloxacin, and nalidixic acid. Several other studies, especially recent ones, have revealed similar high rates of resistance [35,44]. Indeed, the selection and development of antimicrobial-resistant *Campylobacter* are enhanced by the widespread use of these antibiotics in the treatment, management, and disease prevention in livestock.

In the majority of isolates, AR phenotypes corroborate well with the presence of genes and genetic mutations encoding AR. Tetracycline resistance has been linked to the *tet*(O) gene encoding the ribosomal protection protein TetO, which is commonly detected in a variety of Gram-positive and Gram-negative bacteria [45,46]. Furthermore, tetracycline is overused in the avian industry because of its low cost and simplicity of administration through drinking water [47]. It is worth noting that the chicken’s body temperature (42 °C) promotes conjugation and thus contributes to the sharing of plasmids carrying various AR genes [48].

Our isolates showed a high resistance rate to fluoroquinolones. The *cmeABC* operon, encoding multidrug efflux, is the major molecular cause of this resistance in *Campylobacter* [44]. This operon was detected in all our isolates independently of their resistance or susceptibility to quinolones/fluoroquinolones. The second resistance pathway involves one or more point mutations in the QRDR of the GyrA protein, namely the Thr-86-Ile substitution, which is frequently observed in quinolones/fluoroquinolones-resistant isolates [49]. The widespread use of a specific fluoroquinolone (enrofloxacin) in avian industries has caused the selection and wide dissemination of resistant *Campylobacter* strains, which explains the rising resistance trend globally [50]. The world health organization (WHO) classified fluoroquinolones-resistant *Campylobacter* strains as high-priority pathogens resistant to antibiotics, requiring the development of new antibiotics [51,52].

Similarly, numerous studies have shown that the misuse of macrolides in poultry production has resulted in high rates of macrolide resistance in avian *Campylobacter* strains. All erythromycin-resistant *Campylobacter* isolates had the two-point mutations A2075G and/or A2074C in the gene encoding 23S rRNA [49]. The *erm* (B) gene, which can be carried by a variety of multi-drug resistance gene islands (MDRGI), was found in 53.1% [94/177: 48.38% (60/124) *C. jejuni* and 64.15% (34/53) *C. coli*] of our erythromycin-resistant *Campylobacter* isolates. Since the discovery of the *erm*(B) gene in *Campylobacter* in China, it has also been detected in turkey isolates in Spain in 2016 [53] and in the United States in 2016 in a human who previously visited Malaysia [54]. Being found on MDRGIs alongside resistance genes to other antimicrobials including ampicillin, ciprofloxacin, and tetracycline makes noteworthy the presence of this gene in our *Campylobacter* isolates [6]. Since macrolides, in particular erythromycin and azithromycin, are the preferred antibiotics for treating human *Campylobacter* infections, these findings are worrisome.

*Campylobacter* spp. is intrinsically resistant to beta-lactam antibiotics, including ampicillin [55]. However, acquired resistance has been reported. Indeed, enzymatic inactivation by the beta-lactamase encoding gene *bla*_OXA-61_, detected in 18.82% and 6.25% of β-lactam resistant *C jejuni* and *C coli* isolates, respectively, is the main mechanism of acquired ampicillin resistance; in addition, other molecular mechanisms such as porins and the reduced affinity of penicillin-binding protein (PBP) have also been reported [55,56]. The majority of our isolates were gentamicin-susceptible, which is in agreement with previous reports [57,58,59]. This might be linked to its limited use for systemic infections [60], and it is not used in poultry production [58].

### 4.2. Virulence Power of Campylobacter Isolates

The virulome of the *Campylobacter* species contributes to their pathogenicity [61], hence the virulence factors of avian *Campylobacter* need to be investigated for consumer safety. All our isolates had the *flaA*, *cadF*, and *ciaB* genes, which are related to adhesion, colonization, and invasion, as well as the *cdtA*, *cdtB*, and *cdtC* genes, which are critical for CDT expression. The detected frequencies of these genes were analogous to those reported previously from Korea [62], Poland [63], and Italy [64], but higher than those reported from South Africa and Chile [65,66]. The presence of the *cadF* and *ciaB* genes promotes *Campylobacter* adhesion and internalization in cell models [47,67]. The *pldA* gene encoding the outer membrane phospholipase A was detected at a higher rate in *C. jejuni* than in *C. coli*, which is consistent with findings from South Africa [59], Japan [68], and Iran [69]. In addition, regardless of species, all *Campylobacter* isolates contained the *virB*, *racR*, and *dnaJ* genes.

### 4.3. Relationship between Virulence Genes and Phenotypic and Genotypic Antimicrobial Resistance

We identified a possible link between virulence genes and antibiotic resistance by analyzing the antibiotics to which *Campylobacter* isolates are more resistant or susceptible. Interestingly, using Pearson’s chi-square and Fisher’s exact tests, the virulence genes *racR*, *pldA*, *CeuE*, and *cgtB* were found to be closely associated with MDR *Campylobacter* isolates. The same analysis was also performed for each species. *C. jejuni* isolates showed a significant relationship between AR and the different virulence genes, specifically for *racR*, *virB11*, *pldA*, and *cgtB*. The *racR*, *pldA*, and *virB11* genes facilitate bacterial adhesion and intracellular invasion [63]. In addition to the above-mentioned virulence genes linked to *Campylobacter* adhesion and invasion, the *ceuE* gene is one of the four most significant predictor genes in resistant *Campylobacter* isolates. Interestingly, the *cgtB* gene is also a significant gene that has demonstrated a substantial correlation between overall antibiotic resistance status and the prevalence of virulence genes. It is thought to play an important role in the manifestation of Guillain-Barré syndrome, the most severe side effect of human *Campylobacter* infection [70,71]. Since the *cgtB* gene enables bacteria to survive certain stressors, it can also be predicted to be associated with increased AR. However, no significant relationship was observed for *C. coli*, correlating with previous findings [36].

The co-occurrence network demonstrated three distinct networks that illustrate the links between phenotypic AR and the presence or absence of certain virulence genes in each isolate. We observed the coexistence of certain connections between AR and specific virulence genes among the isolates more frequently than others when we focused only on the virulence genes that showed a statistically significant association. There was a high frequency of connections linking nalidixic acid, tetracycline, erythromycin, ciprofloxacin, ampicillin, and chloramphenicol resistance with the virulence genes *pldA* and *racR* in nearly one-third of isolates (*n* = 50). Similarly, when the networks for resistant isolates (*n* = 100) and the total isolates (*n* = 177) were examined, the same high-frequency connections were observed between phenotypic AR and the virulence genes (*pldA* and *racR*).

When we looked at the relationship between virulence genes and AR using various approaches, we noticed that our network visualization matches the Random Forest analysis. We used the Random Forest approach to forecast the value of each virulence gene in order to figure out which gene is more significant for increasing the likelihood of phenotypic resistance in *Campylobacter* isolates. The virulence genes *racR* and *ceuE* were revealed to be the most important predictors of phenotypic resistance in *Campylobacter*. Finally, only one antibiotic, ampicillin, has proven significant value for the *pldA* gene.

## 5. Conclusions

Using statistical and computational tools, we demonstrated the relationship between the distribution of bacterial virulence genes and their phenotypic AR pattern and AR genes among *Campylobacter* isolates from layer hens and eggs. Furthermore, we have shown that the virulence genes *racR*, *pldA*, *virB11*, *ceuE*, and *cgtB* and the AR genes *tet*(O), *cmeB*, and *bla*_OXA-61_, as well as mutations in rRNA 23S and *gyrA*, warrant further investigation using a wide range of antimicrobials to prove links that may increase virulence in bacteria. The findings of this study will be valuable in determining the association between phenotypic traits and genetic characteristics such as the status of virulence and AR genes. Our findings thus open up the possibility for further research into the pathophysiology and the underlying causes of antibiotic resistance.

## Figures and Tables

**Figure 1 foods-11-03554-f001:**
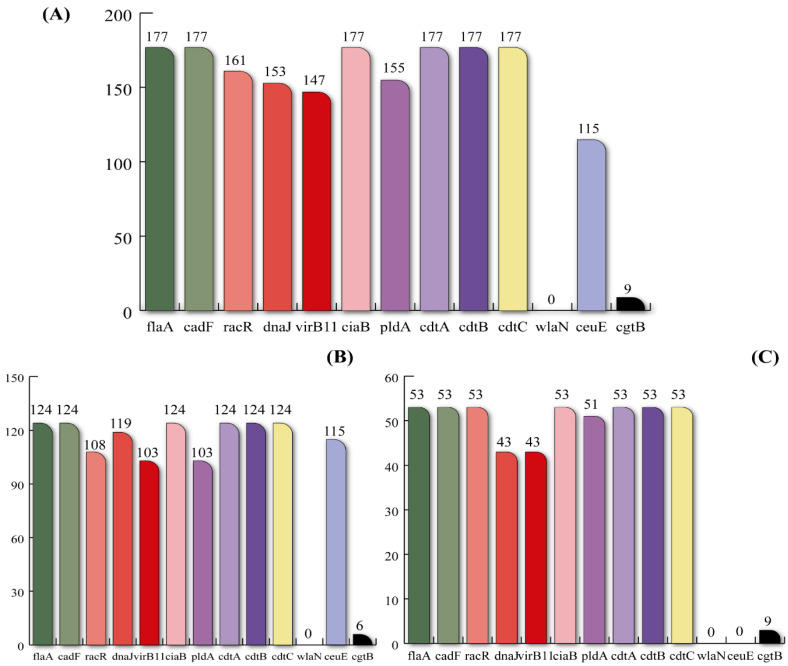
Occurrence of virulence genes (number of positive isolates) across *Campylobacter* isolates (*n* = 177). (**A**) total isolates, (**B**) *C. jejuni* (*n* = 124), (**C**) *C. coli* (*n* = 53).

**Figure 2 foods-11-03554-f002:**
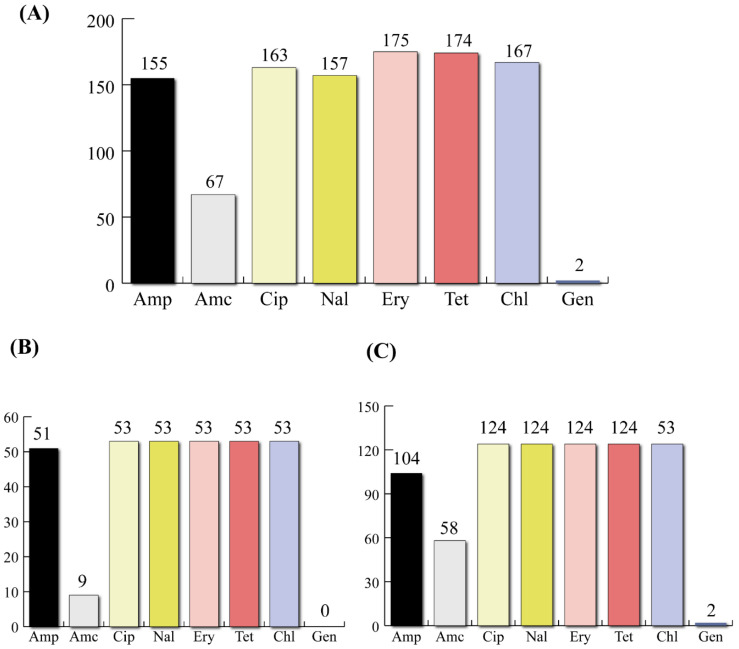
Antibiotic resistance distribution (number of resistant isolates) across *Campylobacter* isolates (*n* = 177). Antibiotic resistance distribution in all *Campylobacter* isolates (**A**), in *C. coli* (*n* = 53) (**B**), and in *C. jejuni* (*n* = 124) (**C**).

**Figure 3 foods-11-03554-f003:**
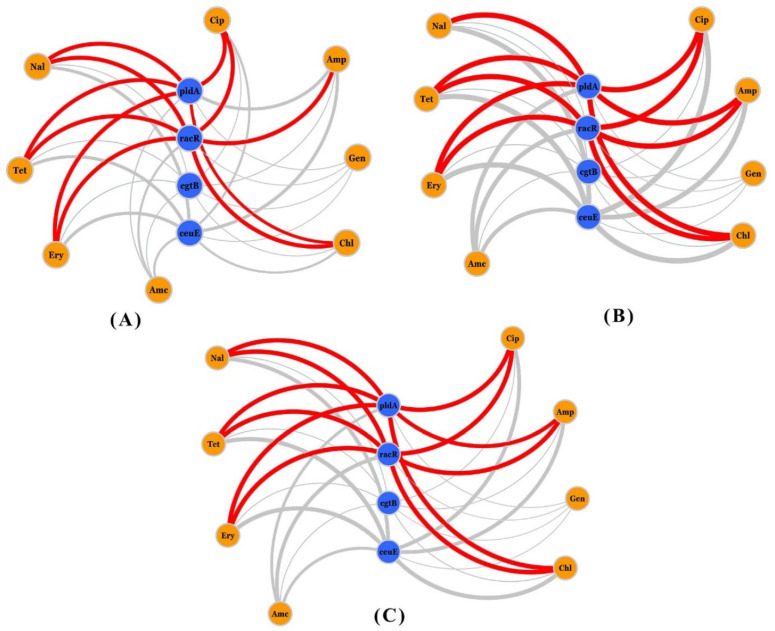
Visualization of the co-occurrence pattern of phenotypic AR and virulence genes across *Campylobacter* isolates (only virulence genes that showed a statistically significant association were used). Red lines indicate a high incidence of links between AR and virulence among isolates. The line thickness between the nodes reveals the frequency of isolates with identical coincident connections. Nodes in orange and blue are AR and virulence genes, respectively. (**A**) Connections between phenotypic AR and virulence genes across 50 *Campylobacter* isolates out of 177 isolates. (**B**) Connections across 100 *Campylobacter* isolates out of 177 isolates. (**C**) Connections across all *Campylobacter* isolates (*n* = 177). Amp, ampicillin; Amc, amoxicillin/clavulanic acid; Cip, ciprofloxacin; Nal, nalidixic acid; Ery, erythromycin; Tet, tetracycline; Gen, gentamicin, and Chl, chloramphenicol.

**Figure 4 foods-11-03554-f004:**
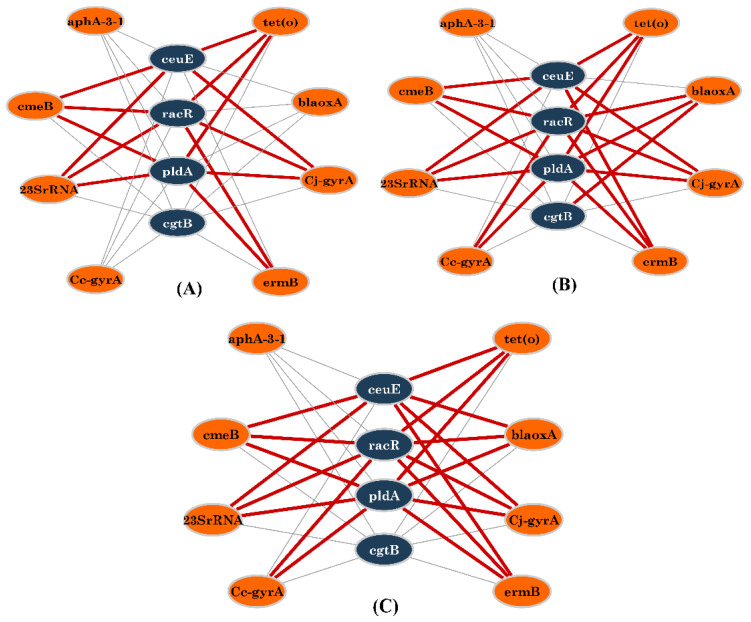
Visualization of the co-occurrence patterns of AR genes and virulence genes across *Campylobacter* isolates (only virulence genes that showed a statistically significant association were used). Red lines designate a high incidence of connections occurring together between resistance genes and virulence among isolates. The line thickness between the nodes reveals the frequency of isolates with identical coincident connections. The nodes in orange and blue are the resistance genes and the virulence genes, respectively. (**A**) Connections across 50 *Campylobacter* isolates out of 177 isolates. (**B**) Connections across 100 *Campylobacter* isolates out of 177 isolates. (**C**) Connections across all *Campylobacter* isolates (*n* = 177). High-frequency connections are shown in red bold lines. AR encoding genes: Quinolones (*gyrA*), erythromycin (23S rRNA), β-lactams (*bla*_OXA-61_), tetracycline (*tet*(O)), and multidrug-resistance pump (*cmeB*).

**Figure 5 foods-11-03554-f005:**
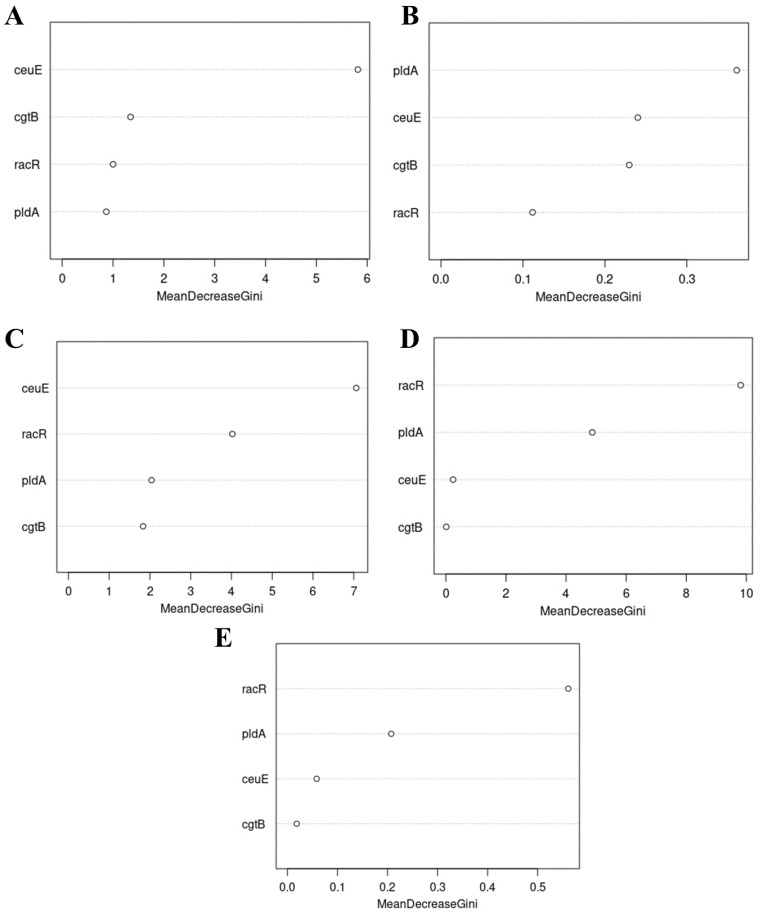
Random Forest analysis displaying the relationship between virulence genes and resistance status for each antibiotic (*n* = 177). Plots display the predominant genes determining the resistance phenotypes through the MeanDecreaseGini value. Prediction of predominant virulence genes for all isolates that have a resistance to a specific antibiotic was observed as follows: (**A**) Amoxicillin, (**B**) Ampicillin, (**C**) Nalidixic Acid, (**D**) Ciprofloxacin, (**E**) Erythromycin, (**F**) Tetracycline, (**G**) Chloramphenicol, (**H**) Gentamicin.

**Table 1 foods-11-03554-t001:** Associations between virulence genes and multidrug resistance in *Campylobacter* isolates (*n* = 177).

Virulence Gene	Absence/Presence *	4–6 Drug Resistance *n* (%)	>6 Drug Resistance *n* (%)	Chi-sq Value	*p* Value (Chi-sq/Fischer)
*flaA*	0	0 (0)	0 (0)	NaN	1
	1	111 (62.71)	66 (37.28)		
*cadF*	0	0 (0)	0 (0)	NaN	1
	1	111 (62.71)	66 (37.28)		
*racR*	0	0 (0)	16 (9.03)	20.379	6.353 × 10^−6^
	1	95 (53.67)	66 (37.28)		
*dnaJ*	0	13 (7.34)	11 (6.21)	0.64529	0.4218
	1	96 (54.23)	57 (32.20)		
*virB11*	0	18 (10.16)	12 (6.8)	1.3439	0.2463
	1	104 (58.75)	43 (24.3)		
*ciaB*	0	0 (0)	0 (0)	NaN	1
	1	110 (62.14%)	67 (37.85%)		
*pldA*	0	62(35.03)	9(5.085)	30.712	2.99 × 10^−8^
	1	49(27.68)	57(32.20)		
*cdtA*	0	0 (0)	0 (0)	NaN	1
	1	111 (62.71)	66 (37.28)		
*cdtB*	0	0 (0)	0 (0)	NaN	1
	1	112 (63.27)	65 (36.72)		
*cdtC*	0	0 (0)	0 (0)	NaN	1
	1	111 (62.71)	66 (37.28)		
*wlaN*	0	111 (62.71)	66 (37.28)	NaN	1
	1	0 (0)	0 (0)		
*ceuE(c,j)*	0	54 (30.50)	8 (4.51)	24.265	8.393 × 10^−7^
	1	57 (32.20)	58 (32.76)		
*cgtB*	0	108 (61.02)	60 (33.9)	6.4249	0.02778
	1	2 (1.13)	7 (3.95)		

*:Absence = 0, Presence = 1; NaN: Not a number.

**Table 2 foods-11-03554-t002:** Associations between virulence genes and multidrug resistance in *C. jejuni* isolates (*n* = 124).

Virulence Gene	Absence/Presence *	4–6 Drug Resistance *n* (%)	>6 Drug Resistance *n* (%)	Chi-sq Value	*p* Value (Chi-sq/Fischer)
*flaA*	0	0 (0)	0 (0)	NaN	1
	1	66 (53.22)	58 (46.77%)		
*cadF*	0	0 (0)	0 (0)	NaN	1
	1	66 (53.22)	58 (46.77)		
*racR*	0	16 (12.9)	0 (0)	16.144	5.871 × 10^−5^
	1	50 (40.32)	58 (46.77)		
*DnaJ*	0	11 (8.87)	4 (3.22)	2.9925	0.1025
	1	54 (43.54)	55 (44.35)		
*virB11*	0	17 (13.70)	4 (3.22)	8.2523	0.004213
	1	48 (38.70)	55 (44.35)		
*ciaB*	0	0 (0)	0 (0)	NaN	1
	1	66 (53.22)	58 (46.77)		
*pldA*	0	18 (14.5)	3 (2.41)	10.718	0.001369
	1	48 (38.70)	55 (44.35)		
*cdtA*	0	0 (0)	0 (0%)	NaN	1
	1	66 (53.22)	58 (46.77)		
*cdtB*	0	0 (0)	0 (0)	NaN	1
	1	66 (53.22)	58 (46.77)		
*cdtC*	0	0 (0)	0 (0)	NaN	1
	1	66 (53.22)	58 (46.77)		
*wlaN*	0	66 (53.22)	58 (46.77)	NaN	1
	1	0 (0)	0 (0)		
*ceuE(c,j)*	0	9 (7.25)	0 (0)	NaN	1
	1	57 (45.96)	58 (46.77)		
*cgtB*	0	66 (53.22)	52 (41.93)	3.5443	0.0933
	1	1 (0.80)	5 (4.03)		

*: Absence = 0, Presence = 1; NaN: Not a number.

**Table 3 foods-11-03554-t003:** Associations between virulence genes and multidrug resistance in *C. coli* isolates (*n* = 53).

Virulence Gene	Absence/Presence *	4–6 Drug Resistance *n* (%)	>6 Drug Resistance *n* (%)	Chi-sq Value	*p* Value (Chi-sq/Fischer)
*flaA*	0	0 (0)	0 (0)	NaN	1
	1	45 (84.9)	8 (15.09)		
*cadF*	0	0 (0)	0 (0)	NaN	1
	1	45 (84.9)	8 (15.09)		
*racR*	0	0 (0)	0 (0)	NaN	1
	1	45 (84.9)	8 (15.09)		
*dnaJ*	0	0 (0)	8 (15.09)	NaN	1
	1	45 (84.9)	0 (0)		
*virB11*	0	0 (0)	8 (15.09)	NaN	1
	1	45 (84.9)	0 (0)		
*ciaB*	0	0 (0)	0 (0)	NaN	1
	1	45 (84.9)	8 (15.09)		
*pldA*	0	0 (0)	1 (1.88)	5.7332	0.1509
	1	45 (84.9)	7 (13.20)		
*cdtA*	0	0 (0)	0 (0)	NaN	1
	1	45 (84.9)	8 (15.09)		
*cdtB*	0	0 (0)	0 (0)	NaN	1
	1	45 (84.9)	8 (15.09)		
*cdtC*	0	0 (0)	0 (0)	NaN	1
	1	45 (84.9)	8 (15.09)		
*wlaN*	0	45 (84.9)	8 (15.09)	NaN	1
	1	0 (0)	0 (0)		
*ceuE(c,j)*	0	45 (84.9)	8 (15.09)	NaN	1
	1	0 (0)	0 (0)		
*cgtB*	0	44 (83.09)	6 (11.32)	6.5995	0.05618
	1	1 ((1.88)	2 (3.77)		

* Absence = 0; Presence = 1; NaN: Not a number.

## Data Availability

The datasets generated and/or analyzed during the current study are available from the corresponding author on reasonable request.

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
