# Peer review of "Relationships between Virulence Genes and Antibiotic Resistance Phenotypes/Genotypes in Campylobacter spp. Isolated from Layer Hens and Eggs in the North of Tunisia: Statistical and Computational Insights"

_foods, 2022, doi:10.3390/foods11223554_

Round 1

Reviewer 1 Report

1) There are many methods for machine learning, why to choose random forest?

2) Why are few reports of human campylobacteriosis detected in North African countries? Really lower level for this disease or under-detection? please analyze this possible reason.

3)  Statistical analysis of phenotypic AR needs the method in detail.

1) Suggest to divide the Discussion into different sub-parts with proper title.

2) L112 Where are these guidances? Need reference source.

Author Response

1) There are many methods for machine learning. Why choose random forest?

Response: Thanks to the reviewer for this comment. There are indeed a lot of machine learning methods out there that are commonly used. In that regard, Tree-based approaches are frequently employed as a kind of bridge between a lot of straightforward models, (logistic regression), that treat characteristics from an independent aspect and other much more intricate models, yet difficult-to-understand from an interpretation point of view.

A random forest is a machine learning approach that is based on decision trees, combining a number of classifiers. The fundamental distinction between the random forest method and other decision trees algorithms is that it randomly selects the root nodes before grouping them, which results, through this system of various decision trees, in a higher prediction accuracy level.

2) Why are few reports of human campylobacteriosis detected in North African countries? Really lower level for this disease or under-detection? please analyze this possible reason.

Response: The majorities of reports on Campylobacter are from poultry or poultry by-products in the context of scientific academic research. However, in human medicine most laboratories in our country and in developing countries in general do not routinely perform tests to detect Campylobacter infection because of the difficulty of the detection methods, especially the growth requirements that need specific and expensive incubation conditions. In addition, molecular methods are very expensive and need qualified laboratory persons.

3)  Statistical analysis of phenotypic AR needs the method in detail.

Response: We thank the reviewer for this comment. As far as we understood the comment, the statistical analysis was explained in the methods section, where we used Fisher exact - test and Pearson’s chi-square to run the association tests. See in section 2.6 : We studied the antimicrobial susceptibility phenotypes (resistance/susceptibility) against the eight tested antibiotics (Amp, Amc, Cip, Nal, Ery, Tet, Chl, and Gen), and associated the later with the presence/absence of investigated virulence genes (cadF, ciaB, racR, flaA, dnaJ, cdtA, cdtB, cdtC, virB11, pldA, wlaN, ceuE, and cgtB). This was investigated first for or all Campylobacter isolates then for isolates of each species. The association test of each virulence gene with the AR phenotype was computed by Pearson’s chi-square or Fisher’s exact test using R software via RStudio (version 1.4.1103). Fisher’s exact test was used when the expected cell counts for the contingency table held less than five isolates. If the p-value < 0.05, the association was deemed statistically significant.

1) Suggest to divide the Discussion into different sub-parts with proper title.

Response: Discussion was divided to 3 sub parts (Antimicrobial resistance and corresponding genotypes// Virulence power of Campylobacter isolates// Relationship between virulence genes and phenotypic and genotypic antimicrobial resistance.

2) L112 Where are these guidances? Need reference source.

Response: We added the reference and the sentence was modified as follows:’….to internationally recognized guidelines ISO 10272-1:2006 (Annex E) for the detection of Campylobacter spp. [33].’ The reference 33 is : ISO 10272-1 (Annex E). 2006. Microbiology of Food And Animal Feeding Stuffs: Horizontal Method for Detection and Enumera-tion of Campylobacter spp. I. Detection Method. International Organization for Standardization, Geneva, Switzerland.

Reviewer 2 Report

The content of this paper is correct. The manuscript is wel written and interesting. The research methodology need to be improved:

 - what was the control strains used for all the analysis?

The manuscript is avery interesting way to look into relationship between the distribution of bacterial virulence genes and their phenotypic AR patter.

Author Response

Dear Reviewer, thank you for your positive evaluation of our work

-Question:  what was the control strains used for all the analysis?

Response: we used well-characterized strains as positive controls strains in every PCR experiments. And we added the following sentence in lines 130 and 141: Positive controls strains from our collection were used in every PCR analysis [36].

Reviewer 3 Report

All comments can be found in the attached PDF.

Author Response

1.Why were no cephalosporins or carbapenems used in the screening? Are these antibiotics of little relevance in Campylobacter treatment?

Response: The treatment of campylobacteriosis is based mainly on the use of quinolones (e.g., ciprofloxacin), macrolides (e.g., erythromycin), and tetracyclines (García-Sánchez, L., Melero, B., & Rovira, J. (2018). Campylobacter in the Food Chain. Advances in food and nutrition research86, 215–252. https://doi.org/10.1016/bs.afnr.2018.04.005.) That is why we do not investigated the antibiotic susceptibility of cephalosporins or carbapenems; In addition in the majority of published articles both antibiotic are rarely used or never tested at all. See also this reference: WHO . Integrated Surveillance of Antimicrobial Resistance in Foodborne Bacteria: Application of a One Health Approach. WHO; Geneva, Switzerland: 2017

  1. Stay consistent in labeling the circle (panel A only shows number of isolates, panels B/C show gene name and number in brackets)

Response: Thanks to the reviewer for this thoughtful comment. We changed the labeling to keep it consistent.

3.General comment: I would consider a bar plot instead of these pies, since the pie suggests that the single slices add up to 100%. Instead, here the sum of the single slices is not a meaningful measure. Finally, something is wrong in panel C / cgtB, the slice is very big but only 3 isolates were positive.

Response: We agree with the reviewer, we changed the donut plot to bar plots.

4.- Same as above, better make a bar plot out of it

- use only one legend for all panels and take the same colors for one antibiotic in all panels (e.g., Amp is once black and then dark blue)

Response: We agree with the reviewer, we changed the donut plots to bar plots and we used the same colors for one antibiotic.

  1. [line 217] Why this cutoff at 6 resistances, is it arbitrary? Else I would wish for a short explanation here. Also, defining > 6 as multidrug resistant seems arbitrary to me, since >=3 classes is already MDR in the classical definition.

Response: We thank the reviewer for this comment. The strains we used for this study are in fact already multiresistant. Because the majority of the strains exhibit at least four drug resistances, we set the cutoff of [4-6] and [>6].

  1. [line 223-224] why is this significant, I thought the cutoff of 0.05 was chosen

Response: We agree with the reviewer for this comment. Indeed, the p-value of racR was incorrectly typed and has been corrected as [χ2 = 16.144, p = 5.871e-05]. We also removed cgtB [χ2 = 3.5443, p = 0.0933] from this paragraph since the p-value is indeed > 0.05.

  1. [table1] It seems that, since we are looking always at the same 177 isolates, the number of isolates in categories 4-6 DRs or >6 DRs ... should be always the same for all virulence genes (e.g. you have 111 of 177 that have 4-6DRs and 66 of 177 that have >6DRs). However, when calculating the sums for the different virulence genes for e.g. category >6DRs, I get sometimes 66, then 82 (e.g. in racR), then 67... How can that be?

Response: We thank the reviewer for this comment. When considering the values in table 1, we look at the 2 lines for 0 (absence of resistance) and 1 (presence of resistance). For each gene, the sum of the  4-6 DRs or the >6 DRs differs from another. When we have 111 4-6 DRs  and 66 >6 DRs for some genes it means in this case that we have 0 sensitive strains for  4-6 DRs and >6 DRs in that particular gene. To take a concrete example, we’ll look at the racR gene mentioned in the reviewer’s comment. In this case, the reviewer mentioned that the sum was 82, and this is indeed the case. When we look at the racR gene we have 66 strains that are sensitive to >6 drugs and 16 that are resistant to >6 drugs , the sum being indeed 82. This means that there are (as indicated in table 1) 0 sensitive strains to  4-6 drugs and 95 strains that are resistant to  4-6 antibiotics. The sum of these 95 strains and the other 82 strains is the total 177 strains.

  1. [table 1 / gene pldA] something must be wrong here, it's the only set where the sum does not equal 177 (17+117+5+60=199)

Response: We agree with the reviewer for this comment. There was indeed a mistake that has been corrected in the manuscript. Table 1 pldA, 17 (9.6) is now 62(35.03), 117(66.10) is now 49(27.68), 5(2.82) is now 9 (5.085) and 60(33.9) is now 57 (32.20), the total is now 177. Chi-square Value is now 30.712 and the p Value (Chi-sq/Fisher) is now 2.99e-08.               

  1. [table 3 /gene cgtB] again, the sum is not =54, there's some mistake here

Response: We agree with the reviewer for this comment. There was indeed a mistake that has been corrected in the manuscript. Table 3 cgtB has been corrected, 45 (83.09) (84.9) is now 44  (83.09)and the total is 53.  Chi-square Value is now 6.5995 and the p Value (Chi-sq/Fisher) is now 0.05618.      

  1. [L 243] needs to be reformulated, sounds somehow wrong

Response: this was corrected as follows: Approximately for the third of the isolates (n=50) there was a high frequency of connections linking nalidixic acid (Nal), tetracycline (Tet), erythromycin (Ery), ciprofloxacin (Cip), ampicillin (Amp) and chloramphenicol (Chl) resistance with the virulence genes pldA and racR (Figure 3A).

  1. [L 250] Maybe the rational for using first only 50, then 100 and finally all the isolates could briefly be explained. Also, how were the subgroups defined, random selection of isolates?

Response: We thank the reviewer for this comment. The rationale behind using 50, 100 and all isolates was to demonstrate any potential relationships between the presence or absence of specific virulence genes and phenotypic AR/resistance genes. We sought to track any potential changes in the co-occurrence of specific antibiotic resistance and virulence genes by gradually and randomly increasing the number of isolates.

To ensure unbiased analysis and findings, the subgroups were created randomly.

  1. [L253] how is "high incidence" defined?

Response: We thank the reviewer for this comment. High incidence is defined by the occurrence together between drug resistance and virulence genes among the isolates.

  1. [L257] repetitive, would delete this

Response: This was corrected

  1. [L264] sounds a bit weird to list 23S rRNA among the resistance genes, maybe at least add a "m" for mutated or write mutated in brackets

Response: This was corrected as follows: ‘….genes: cmeB, tet(O), Cj-gyrA and 23S rRNA (mutated) with the virulence

  1. [L285] higher is the significance

Response: this was corrected.

  1. [L286] you got something wrong here, racR has the highest value in Fig. 5C and 5D only, which is Nal and Cip, respectively.

Response: We agree with the reviewer for this comment. There was a mistake in the order of the panels that led to a mistake in the text. This issue has been corrected in this present manuscript and the paragraph is now: [This investigation showed that one virulence gene, racR, displayed the most important value with two antibiotics, nalidixic acid and ciprofloxacin (Figure 5C, D). On the other hand, another gene, ceuE, has shown the most important value with five other antibiotics, Amoxicillin, Erythromycin, Tetracycline, Chloramphenicol, and Gentamicin (Figure 5 A, E, F, G, H). Finally, the pldA gene showed an important value for Ampicillin only (Figure 5B).] that has been corrected in the manuscript.

  1. [Figure 5] panels must be re-ordered to have panel A on top, not panel E.

Response: We thank the reviewer for this comment The document has been updated to reflect a typo in the panel order.

  1. [L401 - 410] This is like a repetition of the results section, not a discussion. Comparisons to other studies should be drawn here or reflections on why these vir. genes are more associated with AR than others.
  2. [L411 - L417] Again, I don't see a real discussion here, just repetition of the results.

Response to points 18 and 19: Dear reviewer there was not other study in Campylobacter using our strategy to study these associations. We followed a very recent study on Salmonella, that is why we can not compare our results to others.